# "There is just too much going on there": Nonverbal communication experiences of autistic adults

Holly Radford[1]⊙*, Bronte Reidinger[2]⊙*, Steven K. Kapp[1], Ashley de Marchena[3]

**1** School of Psychology, Sport and Health Sciences, University of Portsmouth, Portsmouth, United Kingdom, **2** Psychology Department, Rowan University, Glassboro, New Jersey, United States of America, **3** AJ Drexel Autism Institute, Drexel University, Philadelphia, Pennsylvania, United States of America

⊙ These authors contributed equally to this work.
* holly.radford@port.ac.uk (HR); reidin44@rowan.edu (BR)

## Abstract

### Background

Atypical nonverbal communication is required for a diagnosis of autism, yet little is known about how autistic adults use gestures, facial expressions, and other nonverbal behaviours in social interactions, especially from autistic adults' perspectives. The objectives of this study were to understand: (1) autistic adults' experiences of using nonverbal communication in interactions, (2) how nonverbal communication impacts autistic people's lives, and (3) how autistic adults manage nonverbal communication differences.

### Methods

27 threads from the internet discussion forum wrongplanet.net, all containing dialogue focused on nonverbal communication, were subjected to qualitative analysis. Inductive and deductive coding were used to identify excerpts relevant to miscommunication experiences, communication strengths, and compensatory strategies. A total of 362 excerpts were coded (kappa = .79). Coded excerpts were then extracted and examined for themes, using member checking.

### Results

Major themes included: (1) **Cognitive differences** resulting in autistic adults requiring more time and energy to manage nonverbal communication in interactions; (2) Miscommunication related to nonverbal communication is **bilateral**; (3) Nonverbal communication differences can **negatively impact** the lives and wellbeing of autistic adults; (4) autistic adults use a range of **skills and strategies** to manage nonverbal communication; and (5) Autistic adults demonstrate **variability** in the production and interpretation of nonverbal cues.

**Data availability statement:** Given the complexities of ethical use of social media data, data cannot be shared publicly. However, as all data was pulled from a publicly-viewable forum (wrongplanet.net), interested readers can follow the protocol we used to pull data from the same website, which should yield roughly the same qualitative dataset. Our protocol for pulling data is presented in the main text of our study in the section titled "Data source and extraction". The authors also confirm that they did not have any special access privileges to this data that other researchers would not have. In addition, the authors can make the original dataset available upon reasonable request, on the condition that the interested research team will adhere to the authors' stringent approach to protecting the identity of contributors (i.e., an approach that goes beyond what is required by most ethics boards), and that permission from wrongplanet.net has been sought and granted. Contact information for Wrong Planet can be found here: (https://wrongplanet.net/contact/).

**Funding:** Research reported in this publication was supported in part by the National Institute On Deafness And Other Communication Disorders of the National Institutes of Health under Award Number R21DC020547. The content is solely the responsibility of the authors and does not necessarily represent the official views of the National Institutes of Health. The research for this article was part-funded by the Economic and Social Research Council South Coast Doctoral Training Partnership (Grant Number ES/P000673/1).

**Competing interests:** The authors have declared that no competing interests exist.

## Conclusion

Several of our themes, including mutual miscommunication and negative impacts of atypical communication, are consistent with previous qualitative work on communication experiences of autistic adults. The current findings provide new insight into the internal and external factors influencing the nonverbal communication experiences of autistic adults, in particular the cognitive processes involved. We advocate for solutions that shift the responsibility for effective communication onto all members of society. For example, sharing and accepting preferred communication modalities, and checking in about whether a message was received correctly instead of making assumptions.

## Introduction

Unusual nonverbal communication (NVC), such as tone of voice and prosody (the use of musical aspects of speech such as pitch and rhythm), facial expressions, gestures, and eye gaze, is required for a clinical diagnosis of autism spectrum disorder (henceforth, 'autism') [1]. Experimentally, research has demonstrated significant variability in autistic people's quantifiable NVC skills. (Note: to be consistent with terminology preferences of many within the English-speaking autistic community, identity-first (e.g., 'autistic person'), rather than person-first (e.g., 'person with autism') language [2–5], is used throughout the current manuscript.) For example, in terms of facial expressions, some autistic people are more expressive, and some are less expressive, than non-autistic people [6–9]. These differences may have social consequences: as one example, negative evaluations of job interviews are more closely associated with facial displays for autistic compared to non-autistic candidates [10].

The term "nonverbal" is often used to describe the communication of autistic people who speak few or no words, but in this paper we use the term to describe forms of communication such as facial expressions, gestures, tone of voice and posture that may be used alone or alongside speech, and are used by autistic people who do and do not use speech to communicate. Verbal language and NVC are co-expressive; information conveyed nonverbally can sometimes be redundant with that conveyed verbally, and can sometimes present wholly new information [11]. Likewise, NVC can sometimes happen in isolation (for example, a thumbs up or a nod), but much more often occurs in harmony with verbal communication, such that the integration of both modalities is required to interpret meaning. For example, the simple utterance, "look!" conveys a completely different meaning when said with a tone of awe than with a tone of alarm, and likewise conveys a different meaning along with a point toward a double rainbow compared to an incoming frisbee. NVC also serves a wide range of functions beyond transmitting informational content, including grammatical marking, emphasis, and regulation of turn-taking [12].

In NVC a signaller produces nonverbal cues that are perceived by the receiver and interpreted accordingly. There are three points at which communication can be

misinterpreted: The signals may not match the senders intentions; the receiver may not perceive the signals fully; or the receiver may not interpret them as they were intended. When the signaller and receiver have different perceptions and experiences of the world, for example different cultures and communication norms, they may be more likely to misunderstand one another [13]. In a similar way, the *Double Empathy Problem* describes bidirectional misunderstandings between autistic and non-autistic people, which may extend to NVC, as a result of different perceptions and experiences of the world [14]. But more broadly, people who are more similar to each other, or who are more familiar with each other, may be less likely to misinterpret each other's NVC.

Though few, several high-quality case-control studies provide evidence that autistic adults produce *and* comprehend nonverbal cues differently from non-autistic comparisons [15–19]. By using controlled experimental designs aimed at eliciting specific skills, these studies have elucidated differences in gesture production (specifically, gestures co-produced with speech are harder for non-autistic people to interpret [15]), and in comprehension of prosody [17] and facial expressions [16,18]. Autistic adults are more likely to interpret happy faces as neutral, and neutral faces as sad or angry [18], a finding with clear implications for emotional wellbeing.

Autistic adults have reported that their NVC is often misunderstood by others [20–22], despite studies reporting that autistic adults produce emotional prosody and facial expressions that are *easier* to identify than those produced by non-autistic people [7,17,19]. It seems that whilst some facial expressions are more recognisable (e.g., anger), they are also rated as less natural and more intense [7]. Spontaneous facial expressions produced in naturalistic contexts may also be harder to interpret because they may reflect more than one emotional state, compared to those produced in controlled laboratory environments, where only specific individual emotions are evoked.

Communication differences broadly impact quality of life. With respect to healthcare, autistic adults frequently endorse "difficulty communicating with providers" and "can't process information fast enough" as barriers to healthcare access [23]. Furthermore, a recent interview-based study with autistic adults showed far-reaching effects of negative communication experiences, withdrawal from society and feelings of vulnerability [24]. Nevertheless, few studies have documented autistic adults' personal experiences with NVC specifically, or the impacts on autistic adults of NVC differences. One previous study did analyze autistic individuals' experiences with eye contact as documented on an online discussion forum [25]. Researchers found that autistic individuals had adverse experiences while making eye contact and faced challenges sending and receiving nonverbal information via eye contact. This was in addition to difficulties understanding social nuances such as contexts beyond face-to-face or embodied communication. This study also provides evidence that autistic adults' online self-report of their communication skills and needs can provide useful and informative data. It is unknown if experiences of eye contact specifically – especially in autistic people – can be generalised to other forms of NVC.

Recent studies on social interactions between autistic and non-autistic individuals have begun to acknowledge that social difficulties between these two groups occur in part due to differences in social norms, rather than exclusively due to impaired or atypical skills in autistic people alone [14,26,27]. As NVC plays an important role in social interactions, it is important to use this new framework of mutual responsibility to evaluate NVC experiences of autistic adults. For further research on autistic individuals' social interaction experiences see Watts and colleagues [28].

The current study extends previous research on autistic adults' general communication experiences, and the impact of these experiences, to the study of NVC specifically. Autistic community members endorse further research on communication and communication supports, speaking to the impact of the current study [24]. We analyzed online discussion forum threads written by autistic adults to detail this population's experiences with various forms of NVC in their own words. Our goals were (1) to determine if NVC was experienced as an area of concern for autistic adults, (2) to describe the experiences shared and documented on publicly accessible discussion forums by autistic individuals regarding NVC, and (3) to detail the impact of NVC experiences on autistic adults.

## Method

### Researcher positionality

AdM is a non-autistic psychologist whose research focuses on autistic communication. As a graduate student 15 years ago with no personal contact with autistic adults, AdM learned about first-hand autistic experiences by reading Wrong-Planet.net discussions for hours. BR is a non-autistic graduate student who has an autistic brother who inspired her interest in autism in adulthood. AdM and SK connected professionally in 2021 via their involvement in the *Academic Autism Spectrum Partnership in Research and Education* (AASPIRE) and informally discussed autistic communication, leading AdM and BR to invite SK and HR to collaborate on the project. SK is an autistic autism researcher, who draws from a sensory-movement perspective [29,30]. HR is an autistic autism researcher with an interest in NVC and how it relates to the Double Empathy Problem [14].

### Study design

The present study used a descriptive, qualitative research design and thematic analysis. Importantly, this provides important flexibility, allowing contributors' experiences to be the focus and guide the analysis. Additionally, these methods have been well documented and previously used in studies documenting autistic individuals' lived experiences in a variety of contexts, including social interactions [31–33].

### Data source and extraction

The University of the Sciences Institutional Review Board (IRB) classified the proposed study as not human subjects research; see "Ethical Considerations" section below for a detailed description and discussion of our team's additional ethical decision making. We conducted a web-based search using Google to identify websites with publicly-accessible discussion forums intended for autistic people, using the key phrases "Autism Discussion Forum," "ASD Discussion Forum," "Discussion Forum for Autistic Adults," "Autism Forum," and "General Autism Forum," and evaluated the first 2 pages of results for each search for websites meeting the following inclusionary criteria: English language, targets autistic individuals, has a forum or discussion board format, and possesses NVC-related threads. We excluded websites meeting any of the following criteria: news-focused, a broader focus than autism or very narrow scope (e.g., housing issues for autistic individuals), or private (e.g., requiring accounts to create or view posts, clearly stating they were private, or openly requesting that non-autistic individuals and/or researchers not access it).

Four websites met the criteria above; we selected WrongPlanet (wrongplanet.net) as the study's focus due to it having the largest number of results and its ease of navigation. On 12/22/2020, we searched WrongPlanet for relevant discussion threads using the following in Google: site:wrongplanet.net "body language" OR "facial expressions" OR "eye contact" OR "gestures" OR "nonverbal communication" OR "nonverbal cues", producing 4,620 results, ordered by relevance. Links for the first 200 threads that appeared when conducting our search were extracted and copied to a spreadsheet. An additional extraction of threads related to tone of voice was conducted later (4/22/2021), after recognizing that it was often discussed when developing codes (via the first 20 results for site:wrongplanet.net "tone" OR "voice" OR "inflection" OR "intonation"). This resulted in a total of 220 extracted threads. Saturation of themes resulted in no need for additional data extraction, and could not be predicted in advance. Had additional threads been needed to reach saturation, a secondary extraction would have been conducted. Thus, threads included for analyses could have been written anytime between WrongPlanet's founding in 2004 and the study's extraction in 2020 and 2021.

See Fig. 1 for an overview of the data extraction process. Two authors (BR, AdM) independently evaluated the 220 threads for inclusionary/exclusionary criteria. A thread was included if it: a) was written in English, b) was in a discussion thread format (post and replies, with back and forth conversation), and c) included a keyword in the title or the first post of the thread. A thread was excluded if it: 1) was a duplicate, 2) was private/requiring a login, or 3) included a link to an

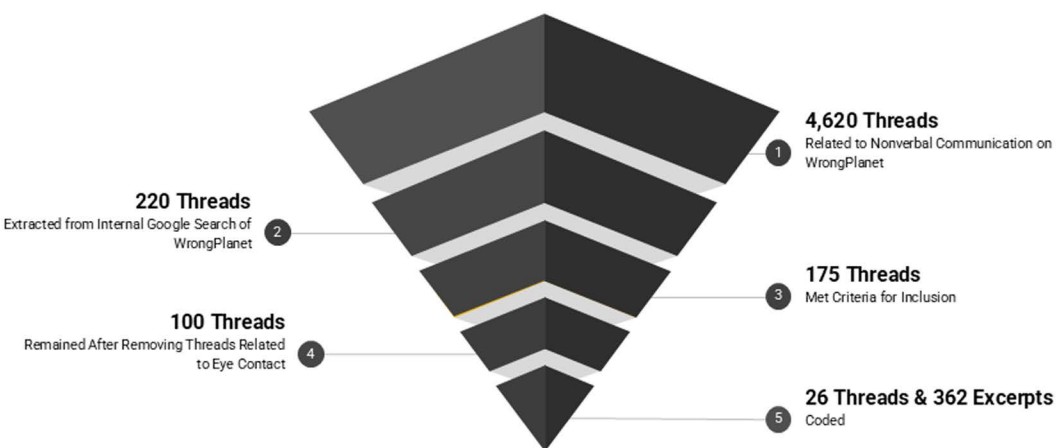

**Fig. 1. Overview of data extraction process.**

outside source in the first post of the thread. 175 threads met criteria to be included, with 95.9% interrater reliability. Disagreements between raters were discussed and resolved.

We note that during coding development, we made the decision to exclude 75 threads focused on eye contact. Eye contact is a critical form of NVC, but we decided to exclude it for two main reasons: (1) a qualitative analysis of WrongPlanet (and YouTube) threads specifically focused on eye contact was recently published [25], and (2) we observed that contributors discussed eye contact differently from other forms of NVC. For example, eye contact was often discussed in the context of social anxiety or sensory aversions, rather than communication.

## Ethical considerations

Using and publishing quotations from publicly available online content raises ethical questions regarding privacy, confidentiality, anonymity, and informed consent. The British Psychological Society Code of Human Research Ethics states that consent must be sought unless observations take place in public, when an individual might reasonably expect to be observed by a stranger [34]. We only analysed chat threads from the publicly-accessible portion of wrongplanet. net in which registration is not required to access the forum or discussion threads. Users who wish to post privately on wrongplanet.net can do so on a private forum which is only accessible to those who have registered and logged in with an account, further supporting the idea by contrast that the posts we analysed were made on a public forum. Our IRB classified this project as non-human subjects research because (1) it did not involve any intervention or interaction with contributors, and (2) the individual identity of users could not be readily ascertained. Wrong Planet granted us permission to use quotes from wrongplanet.net in this paper; data collection and analysis complied with Wrong Planet terms and conditions.

However, we recognise that individual perceptions and expectations of privacy of this discussion forum may vary. Although an online chat forum may be deemed as being in the public domain, this may not be apparent or obvious to users. Understanding of online privacy may also change over time. For these reasons, we took extra precautions to try to protect the privacy and anonymity of contributors. In addition to not reporting usernames, we conducted an internet search to identify whether quotations could be traced back to the original thread or forum. Most quotes could not be traced back to wrongplanet.net, but some quotes could be traced back to the original discussion thread and accounts could be identified. We checked these account profiles to ensure that all usernames were pseudonyms, and that none contained identifying information. We also performed an internet search on all usernames to identify

other accounts on other platforms with the same username that may contain identifying information. A small number of quotations came from usernames which were linked to identifying information on other platforms. To further protect the anonymity of these individuals, we re-wrote these quotations "bricolage style" to make them un-searchable, whilst retaining the original meaning as closely as possible [35]. Paraphrased quotations below are marked with an asterisk.

We note ethical issues regarding the reporting of quotations from online sources without obtaining informed consent. Our team carefully considered the benefits and risks to contributors of whether to report quotations verbatim, or to transfigure them to render them un-searchable. Part of this consideration involved our own positionality, as this issue was first raised by the first author, who is autistic and a user of social media that could potentially be used for research studies such as ours. One benefit of reporting quotations verbatim is that it "gives voice" to the autistic contributors authentically in their own words. We considered this to be a priority, as so often autistic voices are missing in autism research. Reporting quotations verbatim also means that our interpretations as researchers are transparent. However, sharing autistic people's experiences in their own words as much as possible involves a small risk of identification, which we went to great lengths to minimize. We encourage other researchers to take a similar, cautious approach, that goes beyond the current standards of ethics approvals. This is an evolving, nuanced area of research ethics. We hope that sharing our decision process contributes to wider discussions about privacy and anonymity in research using online, user-generated content.

## Data analysis

The final set of threads were coded using inductive-deductive and semantic and latent approaches to thematic analysis, as is commonly used in the literature [36]. Two study authors (AdM and BR) reviewed all threads and selected excerpts relevant to our research question of how autistic adults experience NVC. Excerpts were smaller bits of text from the larger threads that told a personal story, or asked a personal question relevant to NVC. We did not select excerpts in which a contributor described someone else's experiences. We also did not select excerpts from contributors who could reasonably be assumed to be non-autistic based on information they disclosed. The research team used the six phases of reflexive thematic analysis described by Braun and Clarke [37, 38] as a guide throughout the data analysis process, specifically: 1. Familiarization with the dataset, 2. coding, 3. generating initial themes, 4. developing and reviewing themes, 5. refining, defining, and naming themes, and 6. writing up.

We developed codes iteratively; two study authors (BR and AdM) met regularly to discuss and refine the codebook. Once codes were fully developed, the two authors independently coded 28% of excerpts in Dedoose [39]; interrater reliability across codes was kappa = .79 (range:.71 −.93). Discrepancies in coding were discussed until coders reached consensus. Once reliable codes were established, BR evaluated and coded relevant excerpts until thematic saturation was reached, which occurred after coding 26 complete threads. Saturation is defined as the point when no new themes are identified by adding additional data [40]. Consensus meetings were conducted throughout the coding process. 362 excerpts were coded in total.

The two first authors (BR and HR) evaluated all coded excerpts to generate the initial list of themes. Significance for themes was established by our team utilizing a collaborative, subjective, heuristic approach, similar to that described by Maddox and colleagues [41]. Specifically, to be significant, a theme needed to be (1) meaningful for those discussing it, (2) endorsed by multiple contributors, and (3) related to NVC. We note that because verbal and nonverbal communication signals are theorised to arise from a unified system [41], themes for the current study could relate to both verbal and nonverbal communication, though our primary focus was on NVC. Through a series of meetings with all four study authors, we edited, revised, developed, and refined to the initial list of themes to reach the final list of themes detailed below. The two autistic researchers (HR and SKK) also considered whether themes were consistent with their lived experiences and community conversations, as a form of member-checking [42].

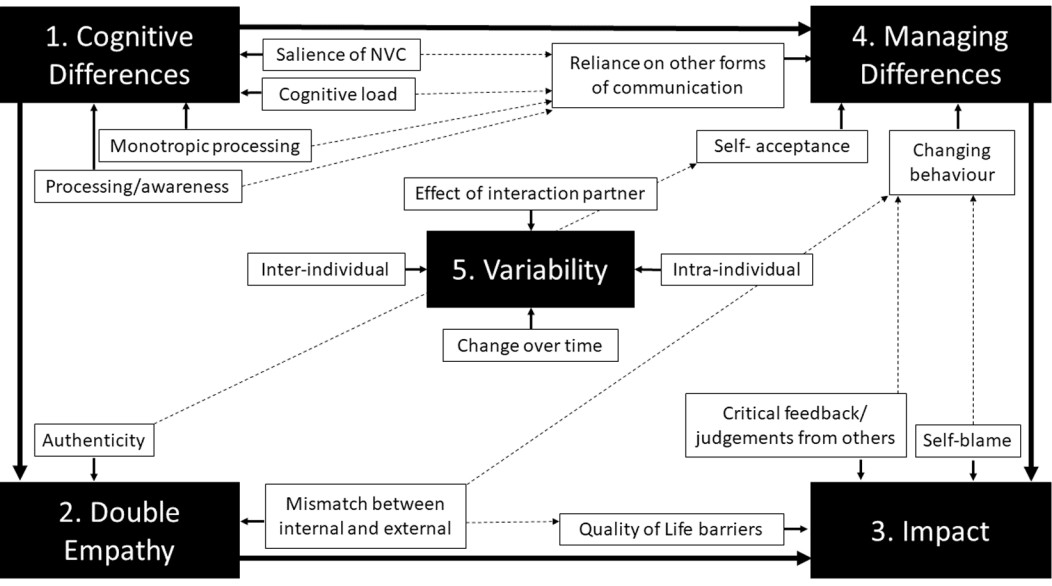

## Results

The authors identified five themes. Themes were highly interrelated and sometimes overlapping. The relationships between the five major themes, and associated subthemes, are depicted in Fig. 2.

### Theme 1: Cognitive differences

A common theme reported by contributors related to differences in cognitive processing of NVC between themselves as autistic individuals, and interaction partners (assumed to be predominantly non-autistic). Four main differences were noted by contributors with respect to cognitive processing of NVC: (1) the *salience* of different forms of communication; (2) *monotropic* processing of communication; (3) processing communication in an *effortful, conscious* way; and (4) cognitive load. We note that these subthemes are highly overlapping as they mutually influence each other during discourse processing.

### Subtheme 1.1 Salience of NVC

Many contributors stated a preference for using verbal and/or written, literal, direct communication, instead of relying on nonverbal cues.

> *"I'm pretty sure that in addition to not being able to read nonverbal cues, I'm just not at all interested in them. They're too open to misinterpretation – too indirect. I've always wished that people would just say what they have to say, and not leave so much unspoken."\**

Reasons for this preference included difficulties in interpreting nonverbal cues, and wanting to avoid being misinterpreted or misinterpreting others.

> *"as long as their facial expressions are written in words with a marker pen on their face. Otherwise they are a maze of furrowed wrinkles that mean nothing but skin contortion in my mind."*

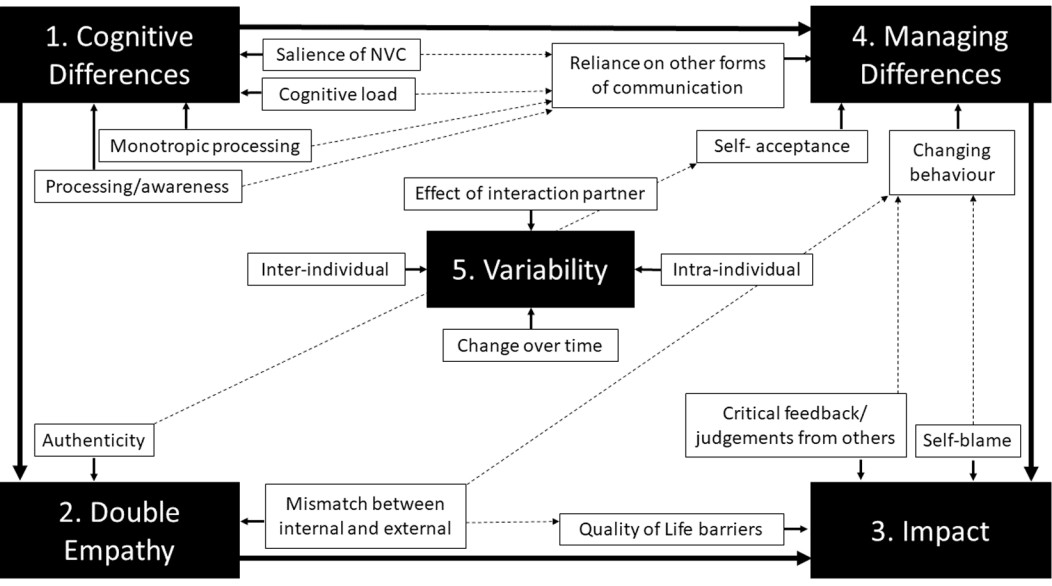

**Fig. 2. Five Major Themes and Associated Subthemes.** Solid arrows depict relationships between major themes and associated subthemes. Dashed arrows highlight relationships between subthemes across major themes that were described by contributors.

Some individuals also reported hardly being aware of nonverbal cues.

*"I do not perceive most non-verbal cues. I hear some changes in vocal tone, and some facial expressions, but very few."*

### Subtheme 1.2 Monotropic processing

Difficulties attending to more than one channel of communication at a time deepened differences in the salience of (non)verbal communication. Many contributors preferred, or found it easier, to attend to one channel of communication over others (e.g., tone of voice) suggesting monotropic processing [43].

*"I spend a good portion of my day on the phone with strangers for my job. Tone now is the only thing I tune into when I'm speaking with someone"*

Trying to process NVC was described by some contributors as "distracting", to the point that they preferred more limited nonverbal expressivity. In other words, processing nonverbal communication (such as prosody) can make it harder to process the content of what is being said.

*"I love monotonous voices, they are easier to listen to […] When someone uses exaggerated/extreme intonation though it becomes very hard for me to follow and it can just be very distracting."*

*"I find myself fascinated by the grand hand gestures some people use. Unfortunately, I have a tendency to study people's moving hands and not hear the words they're saying. My father "talks with his hands" a lot and it can be very distracting."\**

### Subtheme 1.3 Processing/awareness

Contributors mentioned difficulty processing certain aspects of NVC, including understanding the meaning of specific non-verbal cues, knowing how to respond to nonverbal cues, and responding quickly and in the moment to others' nonverbal cues. Some contributors described how a delay in processing NVC could result in being misunderstood.

*"When someone smiles at me it seems I have to notice the smile, process it, think "They're smiling at me; better smile back". By the time that laborious process is through with the other person thinks either I don't like them or I'm in a mood."*

Some stated they rely on previously established knowledge to interpret or know how to respond to nonverbal cues, which is not automatic and takes additional time to process in the moment.

*"To me, reading body language is a conscious process - I have to watch like a hawk for a these little details, then make a conscious effort to correlate them with what I have learned about body language over the course of my life. Looking things up in a "User Manual for Humans", you might say"*

### Subtheme 1.4 Cognitive load

The processing differences outlined by contributors are described as effortful, conscious, time-consuming and tiring, and associated with a higher cognitive load.

*"it's not so much a case of "can/cannot" read body language, so much as a different way of doing it which has a much higher cognitive load, so is much more tiring"*

Comments indicated that the additional burden of processing NVC is the result of having to simultaneously monitor and operate multiple channels of one's own communication and multiple types of communication from others.

*"when I am speaking there are so many annoying things to think about- my voice tone, how much eye contact to make, looking at their body language. there is just too much going on there."*

### Theme 2: The Double Empathy Problem

Contributors ascribed some of their communication difficulties not to deficits within themselves, but to a lack of understanding and flexibility from interaction partners. For example, one contributor said, *"people have told me that reading my facial expressions is like trying to decipher Egyptian hieroglyphics from ancient Greek"*. Many contributors reported that their NVC was frequently misunderstood by others, or even ignored completely:

*"It's ironic that no one can interpret my own tone of voice correctly. My family don't pick up on my nonverbal communication."**

In a few comments, these misunderstandings were attributed to neurotype-specific differences, for example:

*"i do not think that is it true that those with an ASD can not read body language; in most of the gatherings i have been to, everyone knew what the other meant without words, just like in a group of NTs* [neurotypicals]. *In my understanding, those with ASD use a different "type" of body language."*

### Subtheme 2.1: Authenticity

Many contributors stated that they valued genuine, honest, and authentic NVC, and expressed frustration at social pressure to perform inauthentic, or "fake", NVC. An example includes "*My mom always complained that I didn't smile enough-which I thought was ridiculous; why should I walk around smiling if I had nothing in particular to smile about?*"
There was a sense that whilst many people accept, and even expect, that spoken and NVC transmit different messages, that this is not always the case for the autistic interlocutors. In other words, the complex and complementary nature of verbal and nonverbal communication – and what that leaves unspoken – can sometimes come across as inauthenticity to autistic communicators.

*"People…prefer to almost say what they mean and then they have some secret non-verbal thing going on between them. And they assume I can read it too and am playing the same game but I just don't see it and I'm really not communicating any more than I'm speaking"*

### Subtheme 2.2: Mismatch between internal and external state

Some misunderstandings were reported to occur when conversational partners made incorrect inferences about contributors' emotional states, as a result of misinterpreting their nonverbal behaviours:

*"In my teens people used to keep telling me to cheer up when I was in a perfectly good mood."*

Some contributors blamed themselves for these misunderstandings, implying that it must be their own fault for projecting the wrong nonverbal cues, for example "*I must be having wrong expressions on my face. People often seem to get a wrong idea about how I'm feeling because of that".* This also happened conversely; many contributors stated that they make incorrect inferences about others' emotional states: *"think[ing] people are angry at me when they're not".*

### Theme 3: Impact

Negative impacts relevant to NVC were described throughout the forum. Some contributors blamed themselves for breakdowns in communication, whereas others described negative experiences as more reciprocal in nature. Furthermore, negative experiences with NVC led to reduced quality of life for contributors.

### Subtheme 3.1: Self-blame

Many contributors internalised judgments and communication breakdowns they had experienced, placing sole blame and responsibility on themselves, for example, "*I exaggerate my facial expressions now, which sometimes confuses others. Other times my facial expressions show the wrong feeling, or give the wrong impression."**

Some contributors went a step further to express feelings of frustration and anger with themselves for these communication breakdowns. This self-blame sometimes manifested as internalizing the autism deficit narrative, with contributors assuming they were at fault when miscommunications occurred and thus were solely responsible for fixing breakdowns in communication.

> "*I…work hard at my own body language, as it has always played a dominant role in accentuating my loneliness…it used to really make me think 'what the hell is wrong with me?!'"*

### Subtheme 3.2: Critical Feedback/Judgments from Others

Many contributors shared experiences of receiving harsh criticism from others related to their NVC differences. Contributors shared they had been imitated, mocked, "*made fun of*" and accused of being "*odd*" or "*stoned.*" One contributor shared how they were treated while serving in the military: *"in the army I was harassed over my lack of appropriate body language. I was called 'tone-deaf' and 'robotic.'"*

Negative judgments earlier in life continued to influence how some contributors interact – or avoid interaction – with others presently.

> "*Back in high school…usually a comment was made about my reaction. Or the person would laugh because my behavior was strange and unexpected. I'm sure I seemed unfriendly or strange. But I was just confused. To this day, it would still happen if I bothered to go out in public or had male friends.*"

### Subtheme: 3.3 Quality of Life Barriers

Throughout the discussion forum, contributors detailed numerous quality of life barriers – including to social relationships, physical health, and employment – all directly tied to NVC differences, as well as failures of others to accommodate those differences.

Some contributors shared how breakdowns in NVC led to social anxiety.

> "*I believe the social anxiety may have sprouted up so young because I always mistook benign emotions for mean ones. And since I still do that, I often think people are being mean when they're not or don't like me when they do. I also see*

"shocked" looks a lot, when the person apparently is not shocked, so I think something I said has shocked them and get scared. I don't understand it lol."

Others detailed how challenges with NVC affected dating and relationships.

"One area where I struggle in is knowing the difference between a lady who is flirting because she fancies me and a lady who is just being nice. My Mum has said "It was soo obvious! How could you have missed it? She could not have made it any more obvious.""

Additionally, some contributors shared that challenges with NVC affected their ability to make safety assessments.

"I do not perceive most nonverbal cues. This means I have trouble making trustworthy judgements about whether someone is safe or not."

Challenges with NVC also had serious health implications for contributors.

"I've always thought that doctors were angry with me, because they tend to have serious expressions. Plus, since I'm bad at eye contact, they've always suspected I'm lying about my symptoms and medical history. So I've developed such an intense distrust and fear of doctors, that I've avoided seeking necessary medical care, which has had serious repercussions on my physical health."

NVC breakdowns also had serious impacts for contributors' employment.

"At work I'm in a position where I have to try and manage people. This is frequently a disaster and I'm beginning to look at other work options now. The thing is someone can respond negatively or with confusion to something I say or ask of them, but I don't know at the time because I just can't read them at all. First I know of it is when someone tells me I've upset someone."

### Theme 4: Managing differences

Contributors varied in their approaches to managing their NVC differences and the impacts these differences had. They also varied in the success they had with these approaches. Strategies ranged from relying on other forms of communication, to changing their behaviour to meet non-autistic norms, to finding self-acceptance.

### Subtheme 4.1: Reliance on Other Forms of Communication

Contributors frequently mentioned a preference for communication in writing, to avoid the misinterpretations that often resulted from in-person exchanges involving NVC.

"Writing is better because it is just communication stripped to the bare minimum and I can focus my attention without distraction."

Relying on written communication often practically manifested as contributors communicating using online/virtual mediums.

"I don't have to worry about body language, and in message boards and chat rooms the tone is conveyed by obvious-looking smilies."

Some contributors elaborated further, detailing specific situations when reliance on written communication was paramount to social success.

*"I know my shyness has been mistaken for rejection by guys which was why no one would ask me out in the real world so I had to look online for guys because there is no body language."*

### Subtheme 4.2: Changing Behaviour

Some contributors opted to manage their differences by trying to change their nonverbal behaviours to fit societal norms. Contributors varied in the tools they used to try to change their nonverbal behaviour, including taking acting classes, mimicking others, and practicing.

*"In school I had great difficulty with picking up on body language and tones of voice from other people, but as I got older I started to study people's nonverbal communication, and found that I learned it rapidly."**

Some contributors turned to media to help them mold their nonverbal behaviours to fit societal norms.

*"I also studied cartoons and watched lots of commercials and built up a passive/active data bank of patterns of behaviour/expected reaction to a sequence of facial contortions, even if I had a hard time pining down the displayed facial expression."*

Attempts to change behaviour were met with different amounts of success. Some contributors found their attempts were received well.

*"I taught myself to by observing people who used hand gestures. I now do it nearly automatically. I think hand gestures can be a very effective tool for getting your point across."*

In contrast, other contributors found that attempts to change their behaviour were unsuccessful.

*"I've tried very hard to make my voice more 'normal' particularly as I often work in customer service and so it is important not to sound bored. I've spent a long time listening to people speaking on the TV and trying to imitate them. I've managed to make my voice sound less monotonous, but I can't do anything about the pitch and I still sound quite childlike."*

### Subtheme 4.3: Self-Acceptance

Other contributors found focusing on authenticity and self-acceptance to be the best way to manage their differences, rather than attempting to conform to societal norms. This even occurred among contributors who could manage NVC, *"I just outright ignore them, if I offend you tough luck. I do fully understand them though, how to read them and the proper response,"* as well as those who found NVC quite difficult:

*"Since I can't fool them I decided I'd stop trying…There will always be something off and they're always going to be able to tell and it's never going to change your situation so you might as well do what feels natural for you…So instead of looking like a gesticulating idiot I just keep my hands in my pockets or something."*

Some contributors went beyond self-acceptance to exaggerating and integrating their differences into defining characteristics of their personality.

*"What I did to handle this problem, more than struggling to sound different, was to build my style around it. It has a very specific name in English: it's called 'deadpan' and works very well with a serious and monotone (up to a point) voice. Instead of trying to smile and sound lively (I really can't smile) I exaggerated the serious/ironic aspect, usually with a occasional half-smile. People like it."*

### Theme 5: Variability

Many contributors reported that their NVC experiences varied, and that this variation depended sometimes on external and internal factors. *External factor*s included context (who they were interacting with), age and experience, as well as which communication modality was being used. *Internal factors* encompassed variation in their own production or interpretation of different communication modalities. Variability can be seen throughout the other four themes.

### Subtheme 5.1: Inter-individual variability

Differences between contributors were noted. For example, *interpretation* of nonverbal cues ranged from *"I can read people like books"* to *"I have a lot trouble with reading body language"*. Similarly, *production* of nonverbal cues varied from being largely unexpressive (*"I don't know how to properly use facial expressions. The majority of the time, my facial expression is totally blank"*) to quite expressive (*"I use facial expressions…I'm very expressive"*).

### Subtheme 5.2: Intra-individual variability

Some contributors noted a discrepancy between their ability to *express* nonverbal cues and their ability to *interpret* nonverbal cues produced by others. For example, some fared better in interpretation than expression: *"I can definitely understand them, but for me the problem is performing them myself."* Others' expression far exceeded their interpretation: *""i use a lot of hand motions, it helps me get words out faster, but as for body language, i do not understand a tiny bit of body language, it is a big mystery to me"*.

Some reported being hyper-expressive in one modality and hypo-expressive in another, or (as in this example) hyper- and hypo- expressive within the same modality.

*"My voice vacillates between monotone and sing-song. It can't make up its mind about whether to be sub-animated or super-animated, and there is no in-between."*

### Subtheme 5.3: Effect of interaction partner

Some contributors noted that the context of their communication partner influenced how well they could interpret nonverbal cues. Many noted that communication came easier with more familiar people, such as family and friends:

*"With my family, it's pretty simple. I can understand them just fine.…..but in school, I guess people are more confusing"*.

One contributor suggested that this phenomenon resulted from lower stress levels around familiar people.

*"It depends how stressed I am…If I'm relaxed I can have more facial expressions"**

### Subtheme 5.4: Change over time

Contributors' experiences with NVC often had changed over their lives, e.g., through improved understanding:

*"I was terrible at understanding "non-verbal social cues" until I reached about my 30s.I was bad at understanding them till my 40s.I've been "not so bad" at understanding them since---though I do get better with age."*

Some also shared that they were not even aware of NVC until they were older:

*"I'm so bad at reading body language that I didn't know body language even \*existed\* until my teen years"*

A common theme was contributors not picking up on or understanding nonverbal cues as children, but improved awareness and understanding in adulthood:

*"I was in my forties when I realized that I was all wrong when it came to reading body language"*

*"I didn't understand nonverbal cues until I was in my 20s or 30s"\**

## Discussion

We sought to characterise autistic people's lived experiences with NVC via insights derived from naturally occurring, unprompted, chat forum posts and comments – i.e., issues and concerns relevant enough for autistic contributors to spontaneously discuss. We describe five themes that summarise autistic contributors' NVC experiences, specifically: cognitive differences, the Double Empathy problem [14], the impact of these differences and resultant misunderstandings, strategies for managing these differences, and variability of experiences. These together offer insight into the processes involved in producing and interpreting nonverbal cues for autistic adults, and the internal and external factors that contribute both to successful, positive communication experiences, and to miscommunications.

### Cognitive barriers, impacts, and strategies.

Within the theme of cognitive differences, contributors identified difficulty attending to and integrating multiple streams of information simultaneously, a cognitive process especially relevant to NVC. Nonverbals, such as facial expressions and hand gestures – available to the visual system – are often produced at the same time as spoken language – available to the auditory system; these two information streams must be integrated to extract the most comprehensive meaning from our fundamentally multimodal utterances [44,45]. Difficulty with multimodal integration has been robustly demonstrated in autistic people at both a basic perceptual level [46–50], and with respect to communication comprehension and production [46,51–54], and was indeed one of the first cognitive differences identified in autistic children in the 1970s [55,56]. Atypical multimodal integration is likely driven in part by enhanced perceptual functioning in autism [57], which has been demonstrated in both auditory [58] and visual [47] modalities. Autistic people may experience a monotropic style of attention, focused narrowly and deeply on a small area of communication (i.e., one channel/modality), as opposed to a polytropic style of attention which can attend to many channels of communication simultaneously [43].

Many of the strategies for managing NVC differences (Theme 4) appeared to be related to monotropic vs. polytropic attention. For example, nonverbal cues were described as less salient for contributors compared to the content of spoken or written words, and a distinct preference for written communication was apparent, corroborating preferences expressed by autistic adults in previous studies [24,59]. A preference for more uni-modal communication also highlights how some social communication behaviours traditionally conceived of as weaknesses may indeed be strengths in the context of another autistic or accommodating communication partner. For example, one contributor described a *preference* for monotone voices because they found them less distracting.

NVC often came with a heavy *processing cost*. Accordingly, contributors described the *ability* to understand and produce NVC, and integrate it with speech/context, but through deliberate, slow, and effortful processing. Slowed processing

speed has also been robustly demonstrated in autistic people (for a recent meta-analysis, see Zapparrata et al., 2023) [60], including in autistic teens' and adults' responses to verbal and nonverbal communication [53,61–63]. Autistic adults appear to have a general bias toward deliberative processing of information relative to intuitive processing [64], which is consistent with contributors' accounts of how they handle NVC. Once again, accounts from autistic contributors in the current study closely paralleled experimental findings, providing strong support for the validity of autistic self-report of communication skills and experiences. Of note, slowed, deliberative information processing was previously identified as a top barrier to health care by autistic adults [23,65], demonstrating adverse impacts of this processing style when others are unable or unwilling to accommodate it.

**Social barriers, impacts, and strategies.**

Contributors described misinterpreting other people's emotional states and also described that their own internal states were often misinterpreted by others, illustrating a bidirectional communication breakdown. This is consistent with the Double Empathy Problem, whereby differences in perspectives and experiences between people of different neurotypes can result in bidirectional misunderstandings in interactions [14]. It can be assumed that the majority of interaction partners being described were non-autistic, because autistic people tend to default to non-autistic people as co-interactants when questioned about interactions [66]. Indeed, interactions within autistic/autistic dyads are rated more favorably than interactions within autistic/non-autistic dyads [67]. Previous studies have reported that non-autistic people have difficulty understanding the facial expressions and emotional states of autistic people [68,69], and rate autistic prosody more negatively – even when it is easier to understand – supporting the notion that misunderstandings and breakdowns related to NVC are often bi-directional.

Misunderstandings related to NVC had a range of negative impacts for autistic contributors, in areas such as employment, health, and social relationships. For example, many contributors described misinterpreting the facial expressions of others, often in a way that made them feel worse about their interactions (e.g., assuming a doctor was angry at them when because they had a serious facial expression), a finding consistent with experimental work that has found autistic individuals to be more likely to misinterpret neutral faces as angry [16]. Even more striking were experiences of being misunderstood by others, which often happened when an autistic person's words did not "match" their nonverbal expressions, resulting in others not trusting or believing them. Interactions like this with health professionals in particular were described as a barrier to accessing appropriate and timely healthcare, consistent with findings that healthcare providers often ignore autistic patients' communication preferences, leading to health disparities [70,23]. NVC differences' impacts were across NVC modalities: body language, tone of voice, and facial expressions. Contributors described feeling vulnerable, criticised, judged, anxious, and isolated as a result of these difficulties, echoing prior research on autistic adults' communication experiences [24].

Autistic contributors reported using a variety of strategies to manage communication differences, ranging from changing their own behaviour to acceptance of unconventional NVC styles. With respect to changing behaviour, many contributors spoke of investing a significant amount of time and energy observing, studying, and learning about NVC, both formally and informally, in an attempt to ameliorate communication difficulties. However, other contributors felt uncomfortable – and experienced failure anyway – when attempting to use conventional nonverbal cues that felt inauthentic to them. This is consistent with research on strains that autistic adults experience from camouflaging, including exhaustion, poor mental health, and struggle [20,71]. These contributors instead sought to manage NVC differences authentically [72–74], including a strong preference for written forms of communication, which eliminate the need to perform conventional nonverbal cues, supporting similar findings in other studies [24,59]. Self-acceptance was another strategy that reduced the need to behave in inauthentic ways, which guards against depression [75,76]. It involved interacting using contributors' natural communication styles, or emphasizing elements of their authentic communication that were considered socially acceptable, such as dry humour.

## Heterogeneity

Contributors varied dramatically in NVC production and interpretation capabilities. Consistent with contributor reports regarding inter- and intra-individual variation in tone of voice, several previous studies have found that autistic people demonstrate a greater range of pitch variability [17,77,78] than non-autistic people including both increased pitch variability, and decreased pitch variability. One contributor also said that stress influenced NVC abilities, also consistent with previous findings that stress and anxiety further challenge autistic people's ability to communicate in the moment [24,29]. Relatedly, some communication partners facilitated NVC capability, with familiar partners such as family or friends better able to understand, and be understood by, autistic contributors [24]. An additional complexity is that it is not always obvious when someone understands or misunderstands someone else in real time. Interlocutors must *signal* their understanding or lack thereof, an act that often relies on NVC so as not to disrupt the interlocutor's flow of verbal information (e.g., "interactive gestures) [79].

## Recommendations

Consistent with the social model of disability, our findings suggest that support should include developing understanding and skills in society as a whole, not focused solely on developing skills for autistic people. Communication support plans should be individualised, for example the AASPIRE Healthcare Toolkit [80,81], to account for variability of NVC abilities and preference between and within autistic individuals. Our findings support previous recommendations that those interacting with autistic people adapt their communication style, and respect communication preferences [24,59,82], such as using written forms of communication [59]. We also recommend that interaction partners allow sufficient processing time [59,83,84], and slow down the speed of their nonverbal communication [85], as contributors identified a high cognitive load and processing delay when integrating simultaneous channels of NVC. Despite NVC differences, interactions between two autistic people are rated by interlocutors and observers just as positively as interactions between two non-autistic people, and are rated more positively than cross-neurotype interactions [26,27]. This reiterates that many communication difficulties may result from differences between communication styles, rather than deficits within autistic people, suggesting that improving understanding and acceptance among non-autistic people would greatly ameliorate communication difficulties faced by autistic people [86]. We advocate that the responsibility for effective communication be borne by all members of society, and that it should include: checking communication preferences; checking understanding; slowing down the speed of nonverbal communication; being more explicit; and avoiding making assumptions about the meaning of nonverbal cues.

## Limitations

The autistic contributors in this study were able to use internet chat forums, and spontaneously share personal experiences with NVC. Our findings thus do not speak to the NVC experiences of autistic individuals whose amount or type of expressive (written) language output does not allow them to participate in self-reflective discourse about communication. Non-speaking autistic people (some of whom have written language and may be among the contributors) use nonverbal cues such as hand-leading (arguably a form of joint attention to bring another's focus to something in mind [87]) and other bids for attention [88,89], gesturing [90], and reciprocal imitation in response to a communication partner's imitation of their behaviour [88,91]. If neurotypical communication partners do not understand autistic people's communication – including those who are minimally- or non-speaking – autistic people may be excluded from social participation if not broader self-expression. In addition, contributors were limited to those willing to post about their personal experiences with NVC on a public forum.

We know little about the contributors beyond their self-identification as autistic (or on the autism spectrum), including their age, gender, ethnicity or race, and location. The data were collected from 2004–2021, and diagnostic criteria have during that time narrowed somewhat [92], suggesting that the differences participants mentioned are likely to hold up

with time. While participants' autistic status was by self-report (and self-diagnosis was already becoming common during the years of the threads [93]) research supports self-diagnosis in autistic adults for self-report studies such as this one [94–96].

## Future directions

The forums we analyzed were spontaneously generated by autistic users, thus content about NVC was not specifically elicited. Future research using qualitative interviews can strengthen and expand the study's findings by probing NVC specifically and gathering more nuanced information about NVC experiences. We encourage autism researchers using qualitative methods to be thoughtful and creative about making their studies as inclusive and accommodating as possible of those with a range of communication abilities and preferences. For example, our team allows participants to choose their interview format (e.g., phone vs. video conference vs. email/SMS), sends interview questions in advance, and encourages participants to invite a support person; for more examples and guidelines of inclusive research practices, see Nicolaidis and colleagues.

We recognise that there is increasing understanding and acceptance of autism and the neurodiversity movement [95], and as a result the perceptions and experiences described by autistic people may have changed over the time period that we have analysed. It would be useful to compare older and newer forum comments to explore whether autistic people's experiences of NVC have improved over time, in line with increasing levels of understanding and acceptance.

First-hand perspectives on the NVC of minimally or non-speaking autistic people merits further targeted study, such as using Autism Voices, which involves a pre-interview survey and adapted semi-structured interview protocol [97].

## Conclusion

This analysis of spontaneously generated discussion forums amongst autistic adults revealed a number of nuanced insights into the experiences of NVC differences, a defining feature of autism. Cognitive differences described by contributors were largely consistent with the experimental literature, supporting the validity of autistic self-reported communication experiences. Furthermore, contributors described substantial amounts of psychological distress related to communication breakdowns grounded in NVC, which were largely bilateral in nature. As shown in Fig. 2, the five major themes are related in dynamic and complex ways, ultimately resulting in a high degree of practical and emotional impact on outcomes of importance to our contributors. The most commonly given recommendations for interaction partners echo the work of others, namely: allowing sufficient processing time for both comprehension and production, respecting communication preferences (e.g., for written communication), and trusting autistic people's words even when their nonverbals appear to be sending a conflicting signal.

## Acknowledgments

The authors would like to thank WrongPlanet.net contributors and Cornflake (wrongplanet.net administrator) for facilitating permission to use threads for research. We also thank members of the University of the Sciences *InterAction Lab* for their help with data extraction, and members of the Early Detection and Intervention team at the AJ Drexel Autism institute for help with formatting. Portions of this work were presented at the 2022 meeting of the *International Society for Gesture Studies*, and the 2023 meeting of the *International Society for Autism Research.*

## Author contributions

**Conceptualization:** Bronte Reidinger, Ashley de Marchena.

**Formal analysis:** Holly Radford, Bronte Reidinger, Steven K Kapp, Ashley de Marchena.

**Methodology:** Steven K Kapp, Ashley de Marchena.

**Writing – original draft:** Holly Radford, Bronte Reidinger.

**Writing – review & editing:** Steven K Kapp, Ashley de Marchena.

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
