## [Decision Letter · Decision Letter 0]

Thank you for submitting your manuscript to PLOS ONE. After careful consideration, we feel that it has merit but does not fully meet PLOS ONE’s publication criteria as it currently stands. Therefore, we invite you to submit a revised version of the manuscript that addresses the points raised during the review process.

We look forward to receiving your revised manuscript.

Kind regards,

Saeid Sadeghi

Academic Editor

PLOS ONE

**Journal Requirements:**

2. In your Methods section, please include additional information about your dataset and ensure that you have included a statement specifying whether the collection and analysis method complied with the terms and conditions for the source of the data.

Research reported in this publication was supported in part by the National Institute On Deafness And Other Communication Disorders of the National Institutes of Health under Award Number R21DC020547. The content is solely the responsibility of the authors and does not necessarily represent the official views of the National Institutes of Health.

The research for this article was part-funded by the Economic and Social Research Council South Coast Doctoral Training Partnership (Grant Number ES/P000673/1).

4. For studies involving third-party data, we encourage authors to share any data specific to their analyses that they can legally distribute. PLOS recognizes, however, that authors may be using third-party data they do not have the rights to share. When third-party data cannot be publicly shared, authors must provide all information necessary for interested researchers to apply to gain access to the data. (https://journals.plos.org/plosone/s/data-availability#loc-acceptable-data-access-restrictions) 

a) A description of the data set and the third-party source

b) If applicable, verification of permission to use the data set

c) Confirmation of whether the authors received any special privileges in accessing the data that other researchers would not have

d) All necessary contact information others would need to apply to gain access to the data

5. We notice that your supplementary figures are uploaded with the file type 'Figure'. Please amend the file type to 'Supporting Information'. Please ensure that each Supporting Information file has a legend listed in the manuscript after the references list.

**Additional Editor Comments:**

Dear author,

Thank you for submitting your manuscript to PLOS One.  

I have completed my evaluation of your manuscript. The reviewers recommend reconsideration of your manuscript following revision. I invite you to resubmit your manuscript after addressing the comments.

Below, I have summarized the key points from the reviewer’s comments:

1. General Feedback: The reviewer appreciates the modern and cost-effective approach of your study and finds the topic to be very interesting. They found the introduction to be understandable, informative, and pleasant to read.

2. Terminology: The reviewer raised a concern regarding the use of the phrase “autistic adults” in the abstract and introduction. They suggested considering an alternative formulation, such as “adults with autism,” and would appreciate a brief explanation of your choice of terminology.

3. Methods Section: The reviewer suggested ensuring verifiability by potentially making original treats available “on demand” for other researchers. Additionally, they mentioned the possibility of conducting “member checking” with a subgroup of forum members, though they noted that this is not strictly necessary. They also requested a more detailed description of the criteria for saturation mentioned in line 266, as well as a more detailed analysis of how themes were derived from the coding process.

4. Results Section: The reviewer noted two minor points: introducing the abbreviation “NTs” before its first use and ensuring that quotes align with relevant keywords, such as “authenticity.”

5. Discussion Section: The reviewer noted some recent studies that could enhance your discussion. They also highlighted a need for clarity on how non-autistic conversation partners could adapt their communication when interacting with individuals on the spectrum.

I encourage you to carefully consider these comments and make any necessary revisions to your manuscript. Please submit your revised manuscript along with a detailed response to each comment.

Reviewers' comments:

Reviewer's Responses to Questions

**Comments to the Author**

1. Is the manuscript technically sound, and do the data support the conclusions?

Reviewer #1: Yes

2. Has the statistical analysis been performed appropriately and rigorously?

Reviewer #1: N/A

3. Have the authors made all data underlying the findings in their manuscript fully available?

Reviewer #1: No

4. Is the manuscript presented in an intelligible fashion and written in standard English?

Reviewer #1: Yes

**Reviewer #1: ** This study uses a modern and cost-effective approach to obtain first-hand information about a very interesting topic. With minor revisions, I am pleased to recommend this article for publication.

THE INTRODUCTION is understandable, informative and pleasant to read.

What I already noticed in the abstract and even more so in the introduction is the consequent use of phrases such as “autistic adults”. Was this formulation discussed and deliberately chosen? It is of course possible to use it, and, in many cases, this is the preferred self-description of those affected. However, these formulations can also have a stigmatizing effect, especially if the context focuses on the problems and deficits caused by the autistic condition. An alternative formulation could be “adults with autism”, which has more of a connotation of ONE aspect of the personality of those affected, as opposed to autism being the main aspect. Presumably this aspect has already been considered, but I (and possibly also the potential readers of the article) would appreciate a brief explanation of this choice.

THE METHODS section is comprehensible and appears robust. The arguments against publishing the original treats are valid, but in addition to the protection of the investigated persons, verifiability must be ensured. What about making them available “on demand” for other scientists who want to build on them or review them?

At least one option seems necessary to me. Not quite as necessary, but desirable, would be a kind of "member checking" of the results. For example, could an evaluation of the results be carried out with a subgroup of forum members or even openly in the forum? If this is not possible, I think it can be omitted as most of the results are not very controversial.

Also, there is no description of the criteria for saturation mentioned in line 266. The Coding process is described quite well, but the analysis part leading to the themes and their interaction in the results could be described in more detail to make it more reproducible.

THE RESULTS appear lively with many citations and are quite interesting! Even if many of the statements are not very surprising, it is helpful for understanding to read them in such a structured way. The graphic presentation is also well done.

Two small comments: in line 385, “NTs” is used as an abbreviation without “neurotypicals” having been introduced with this abbreviation beforehand.

And the quote from line 392 onwards doesn't really seem to fit the keyword “authenticity”.

THE DISCUSSION comprehensively places the results in the context of current research, but some current studies on the topic do not seem to be discussed. A brief Pubmed search revealed the following studies, which could be added:

1. Thorsson, M., Galazka, M. A., Åsberg Johnels, J., & Hadjikhani, N. (2024). Influence of autistic traits and communication role on eye contact behavior during face-to-face interaction. Scientific reports,14(1), 8162. https://doi.org/10.1038/s41598-024-58701-8

2. Strömberg, M., Liman, L., Bang, P., & Igelström, K. (2022). Experiences of Sensory Overload and Communication Barriers by Autistic Adults in Health Care Settings. Autism in adulthood : challenges and management, 4(1), 66–75. https://doi.org/10.1089/aut.2020.0074

The recommendation that non-autistic conversation partners should also take responsibility for adapting their communication is interesting, but it is not clear how this could be implemented in practice. Especially for people who do not regularly deal with people on the spectrum and know little about them. Here, for example, cards with conversation rules or helpful tips for successful communication could be given to the conversation partners (e.g. please give me more time to answer).

**Do you want your identity to be public for this peer review?** For information about this choice, including consent withdrawal, please see our Privacy Policy

Reviewer #1: **Yes: ** Michael Alexander Pelzl

---

## [Author Response · Author response to Decision Letter 1]

2 Apr 2025

Journal Requirements:

Done.

2. In your Methods section, please include additional information about your dataset and ensure that you have included a statement specifying whether the collection and analysis method complied with the terms and conditions for the source of the data.

This has been added.

Research reported in this publication was supported in part by the National Institute On Deafness And Other Communication Disorders of the National Institutes of Health under Award Number R21DC020547. The content is solely the responsibility of the authors and does not necessarily represent the official views of the National Institutes of Health.

The research for this article was part-funded by the Economic and Social Research Council South Coast Doctoral Training Partnership (Grant Number ES/P000673/1).

The amended Funding Statement is as follows:

“Research reported in this publication was supported in part by the National Institute On Deafness And Other Communication Disorders of the National Institutes of Health under Award Number R21DC020547. The content is solely the responsibility of the authors and does not necessarily represent the official views of the National Institutes of Health. The research for this article was part-funded by the Economic and Social Research Council South Coast Doctoral Training Partnership (Grant Number ES/P000673/1). There was no additional external or internal funding received for this study.”

4. For studies involving third-party data, we encourage authors to share any data specific to their analyses that they can legally distribute. PLOS recognizes, however, that authors may be using third-party data they do not have the rights to share. When third-party data cannot be publicly shared, authors must provide all information necessary for interested researchers to apply to gain access to the data. (https://journals.plos.org/plosone/s/data-availability#loc-acceptable-data-access-restrictions)

a) A description of the data set and the third-party source

b) If applicable, verification of permission to use the data set

c) Confirmation of whether the authors received any special privileges in accessing the data that other researchers would not have

d) All necessary contact information others would need to apply to gain access to the data

We have described our position on data sharing in the response to Reviewer #1, below.

5. We notice that your supplementary figures are uploaded with the file type 'Figure'. Please amend the file type to 'Supporting Information'. Please ensure that each Supporting Information file has a legend listed in the manuscript after the references list.

Done.

Additional Editor Comments:

1. General Feedback: The reviewer appreciates the modern and cost-effective approach of your study and finds the topic to be very interesting. They found the introduction to be understandable, informative, and pleasant to read.

We thank you both for your constructive and positive review!

2. Terminology: The reviewer raised a concern regarding the use of the phrase “autistic adults” in the abstract and introduction. They suggested considering an alternative formulation, such as “adults with autism,” and would appreciate a brief explanation of your choice of terminology.

We have addressed this concern in the paper and in our response to Reviewer #1, below.

3. Methods Section: The reviewer suggested ensuring verifiability by potentially making original treats available “on demand” for other researchers. Additionally, they mentioned the possibility of conducting “member checking” with a subgroup of forum members, though they noted that this is not strictly necessary. They also requested a more detailed description of the criteria for saturation mentioned in line 266, as well as a more detailed analysis of how themes were derived from the coding process.

We have addressed this feedback in response to Reviewer #1 below. We were not completely certain if we needed to add more information about data sharing to the manuscript itself, we are happy to add that information wherever it would be most beneficial.

4. Results Section: The reviewer noted two minor points: introducing the abbreviation “NTs” before its first use and ensuring that quotes align with relevant keywords, such as “authenticity.”

Addressed.

5. Discussion Section: The reviewer noted some recent studies that could enhance your discussion. They also highlighted a need for clarity on how non-autistic conversation partners could adapt their communication when interacting with individuals on the spectrum.

Addressed – we appreciate the suggestions.

Review Comments to the Author

Reviewer #1: This study uses a modern and cost-effective approach to obtain first-hand information about a very interesting topic. With minor revisions, I am pleased to recommend this article for publication.

We appreciate the thoughtful review!

THE INTRODUCTION is understandable, informative and pleasant to read.

What I already noticed in the abstract and even more so in the introduction is the consequent use of phrases such as “autistic adults”. Was this formulation discussed and deliberately chosen? It is of course possible to use it, and, in many cases, this is the preferred self-description of those affected. However, these formulations can also have a stigmatizing effect, especially if the context focuses on the problems and deficits caused by the autistic condition. An alternative formulation could be “adults with autism”, which has more of a connotation of ONE aspect of the personality of those affected, as opposed to autism being the main aspect. Presumably this aspect has already been considered, but I (and possibly also the potential readers of the article) would appreciate a brief explanation of this choice.

We appreciate the reviewer’s concerns about the use of possibly stigmatizing language and welcome the opportunity to clarify our choices, as we acknowledge future readers may have the same concerns. To be consistent with terminology preferences of the English-speaking autistic community as documented in recent research internationally (Bottema-Beutel et al., 2021; Keating et al., 2023; Robertson et al., 2025), we used identity-first (e.g. autistic person), rather than person-first (e.g. person with autism) language, throughout the current manuscript. In these cultures, preference for person-first language is associated with more stigma toward autistic people (Bury et al., 2023). We have added a footnote at the first use of identity-first language (i.e. autistic person) in the introduction to explain this choice.

Robertson, Z. S., Stockwell, K. M., Lampi, A. J., & Jaswal, V. K. (2025). Autism Terminology Preferences Among Autistic and Non-Autistic Adults in North America. Autism in Adulthood.

THE METHODS section is comprehensible and appears robust. The arguments against publishing the original treats are valid, but in addition to the protection of the investigated persons, verifiability must be ensured. What about making them available “on demand” for other scientists who want to build on them or review them?

We are happy to make the original threads available upon request, on the condition that we are satisfied that the research team will adhere to our stringent approach to protecting the identity of our contributors (i.e., an approach that goes beyond what is required by most ethics boards), and that permission from wrongplanet.net has been sought and granted. We are glad you agree that mandatory sharing of qualitative data could be unethical, as it may involve sharing information as sensitive as people’s autism diagnosis and potentially traumatic experiences they have had (Prosser, Bagnall, & Higson-Sweendey, 2024; Prosser et al., 2023). We describe the search process in great detail so that interested readers can identify and reproduce relevant threads on their own.

Prosser AM, Bagnall R, Higson-Sweeney N. Reflection over compliance: Critiquing mandatory data sharing policies for qualitative research. Journal of Health Psychology. 2024 Jun;29(7):653-8.

Prosser AM, Hamshaw RJ, Meyer J, Bagnall R, Blackwood L, Huysamen M, Jordan A, Vasileiou K, Walter Z. When open data closes the door: A critical examination of the past, present and the potential future for open data guidelines in journals. British Journal of Social Psychology. 2023 Oct;62(4):1635-53.

At least one option seems necessary to me. Not quite as necessary, but desirable, would be a kind of "member checking" of the results. For example, could an evaluation of the results be carried out with a subgroup of forum members or even openly in the forum? If this is not possible, I think it can be omitted as most of the results are not very controversial.

We have added a statement in our data analysis sub-section about member-checking: Two autistic members of our research team considered whether themes were consistent with their lived experience and community conversation, as a form of member-checking (Raymaker et al., 2020).

Also, there is no description of the criteria for saturation mentioned in line 266. The Coding process is described quite well, but the analysis part leading to the themes and their interaction in the results could be described in more detail to make it more reproducible.

We have added a definition and citation for the concept of thematic saturation. We have also added some information about the process of identifying and developing themes, and have made explicit the key references that informed our approach, for additional transparency, and for readers who are interested in adopting a similar approach.

THE RESULTS appear lively with many citations and are quite interesting! Even if many of the statements are not very surprising, it is helpful for understanding to read them in such a structured way. The graphic presentation is also well done.

Two small comments: in line 385, “NTs” is used as an abbreviation without “neurotypicals” having been introduced with this abbreviation beforehand.

And the quote from line 392 onwards doesn't really seem to fit the keyword “authenticity”.

We have added the word “neurotypical” in brackets immediately after its use by the contributor. We have also added an explanation of how the quote pointed out by the reviewer relates to the theme of authenticity. Thank you for allowing us to clarify!

THE DISCUSSION comprehensively places the results in the context of current research, but some current studies on the topic do not seem to be discussed. A brief Pubmed search revealed the following studies, which could be added:

1. Thorsson, M., Galazka, M. A., Åsberg Johnels, J., & Hadjikhani, N. (2024). Influence of autistic traits and communication role on eye contact behavior during face-to-face interaction. Scientific reports,14(1), 8162. https://doi.org/10.1038/s41598-024-58701-8

2. Strömberg, M., Liman, L., Bang, P., & Igelström, K. (2022). Experiences of Sensory Overload and Communication Barriers by Autistic Adults in Health Care Settings. Autism in adulthood : challenges and management, 4(1), 66–75. https://doi.org/10.1089/aut.2020.0074

The recommendation that non-autistic conversation partners should also take responsibility for adapting their communication is interesting, but it is not clear how this could be implemented in practice. Especially for people who do not regularly deal with people on the spectrum and know little about them. Here, for example, cards with conversation rules or helpful tips for successful communication could be given to the conversation partners (e.g. please give me more time to answer).

Thank you for the paper recommendations. We felt that the first suggestion was not within scope because it was about autistic traits rather than autistic people. We have included a citation to Strömberg et al., (2022), and have added recommendations about slowing down the speed of nonverbal communication, and being more explicit.

---

## [Decision Letter · Decision Letter 1]

“There is just too much going on there”: Nonverbal communication experiences of autistic adults

PONE-D-24-49426R1

Dear Dr. Reidinger,

We’re pleased to inform you that your manuscript has been judged scientifically suitable for publication and will be formally accepted for publication once it meets all outstanding technical requirements.

Kind regards,

Saeid Sadeghi

Academic Editor

PLOS ONE

Additional Editor Comments (optional):

Thank you for submitting a revised version of your manuscript. Having considered your revisions and responses, we're delighted to let you know that your manuscript has been accepted for publication in PLOS ONE. We wish you continued success in your research endeavors.

Reviewers' comments:

Reviewer's Responses to Questions

**Comments to the Author**

Reviewer #1: All comments have been addressed

2. Is the manuscript technically sound, and do the data support the conclusions?

Reviewer #1: Yes

3. Has the statistical analysis been performed appropriately and rigorously?

Reviewer #1: N/A

4. Have the authors made all data underlying the findings in their manuscript fully available?

Reviewer #1: No

5. Is the manuscript presented in an intelligible fashion and written in standard English?

Reviewer #1: Yes

Reviewer #1: There are no further comments from my side, I thank the authors for their clarifications and additions.

**Do you want your identity to be public for this peer review?** For information about this choice, including consent withdrawal, please see our Privacy Policy

Reviewer #1: **Yes: ** Michael Alexander Pelzl

---

## [Editor Report · Acceptance letter]

PONE-D-24-49426R1

PLOS ONE

Dear Dr. Reidinger,

I'm pleased to inform you that your manuscript has been deemed suitable for publication in PLOS ONE. Congratulations! Your manuscript is now being handed over to our production team.

Kind regards,

on behalf of

Dr. Saeid Sadeghi

Academic Editor

PLOS ONE